# Leaf Age-Dependent Photosystem II Photochemistry and Oxidative Stress Responses to Drought Stress in *Arabidopsis thaliana* Are Modulated by Flavonoid Accumulation

**DOI:** 10.3390/molecules26144157

**Published:** 2021-07-08

**Authors:** Ilektra Sperdouli, Julietta Moustaka, Georgia Ouzounidou, Michael Moustakas

**Affiliations:** 1Department of Botany, Aristotle University of Thessaloniki, 54124 Thessaloniki, Greece; ilektras@bio.auth.gr (I.S.); moustaka@plen.ku.dk (J.M.); 2Institute of Plant Breeding and Genetic Resources, Hellenic Agricultural Organization-Demeter (ELGO-Dimitra), Thermi, 57001 Thessaloniki, Greece; 3Department of Plant and Environmental Sciences, University of Copenhagen, Thorvaldsensvej 40, 1871 Frederiksberg, Denmark; 4Institute of Food Technology, Hellenic Agricultural Organization-Demeter (ELGO-Dimitra), 1 S. Venizelou Str., 14123 Lycovrissi, Greece; geouz@nagref.gr

**Keywords:** acclimation, non-photochemical quenching (NPQ), mild drought stress, moderate drought stress, young leaves, mature leaves, lipid peroxidation, singlet oxygen (^1^O_2_), lipid peroxidation, reactive oxygen species (ROS)

## Abstract

We investigated flavonoid accumulation and lipid peroxidation in young leaves (YL) and mature leaves (ML) of *Arabidopsis thaliana* plants, whose watering stopped 24 h before sampling, characterized as onset of drought stress (OnDS), six days before sampling, characterized as mild drought stress (MiDS), and ten days before sampling, characterized as moderate drought stress (MoDS). The response to drought stress (DS) of photosystem II (PSII) photochemistry, in both leaf types, was evaluated by estimating the allocation of absorbed light to photochemistry (Φ*_PSII_*), to heat dissipation by regulated non-photochemical energy loss (Φ*_NPQ_*) and to non-regulated energy dissipated in PSII (Φ*_NO_*). Young leaves were better protected at MoDS than ML leaves, by having higher concentration of flavonoids that promote acclimation of YL PSII photochemistry to MoDS, showing lower lipid peroxidation and excitation pressure (1 − q_*p*_). Young leaves at MoDS possessed lower 1 − q_*p*_ values and lower excess excitation energy (EXC), not only compared to MoDS ML, but even to MiDS YL. They also possessed a higher capacity to maintain low Φ*_NO_*, suggesting a lower singlet oxygen (^1^O_2_) generation. Our results highlight that leaves of different developmental stage may display different responses to DS, due to differential accumulation of metabolites, and imply that PSII photochemistry in *Arabidopsis thaliana* may not show a dose dependent DS response.

## 1. Introduction

Drought is a major limiting factor for plant growth and crop productivity [1,2] and is expected to increase in intensity, frequency and duration as a consequence of climate change [3,4,5]. Drought stress (DS) accelerates leaf senescence [6,7] and impairs osmotic adjustment of plants and plants′ photosynthetic rate and growth [6,8], thus reducing plant productivity that affects food security [4,9]. Indeed, even a short-term DS results in crucial annual losses of crop yields, preventing sustainable agriculture [5]. Consequently, an understanding of DS in relation to plant growth and development is of importance for sustainable agriculture [10]. 

Plants use multiple strategies to either avoid or tolerate drought stress [7]. Drought tolerance is associated with maintenance of plant water status through osmotic adjustment by the accumulation of osmoprotective substances in leaves, such as soluble sugars and proline, that help the plants to maintain leaf water status [11,12,13,14,15] and to acclimate to DS [14,16]. Drought stress can appear in an extensive range, from mild drought stress (MiDS), to moderate drought stress (MoDS), to severe drought stress (SDS), during which plants experience dehydration and wilting, finally leading to their death [17]. Several earlier studies have concentrated on SDS, though, MoDS is appearing more often in real field conditions [17]. More recent studies have demonstrated that plants apply different strategies to cope with MiDS, compared to MoDS or to SDS [10,18,19,20]. For example, between SDS and MiDS, only one third of differentially expressed genes overlap in Arabidopsis young leaves (YL) [20], while photosynthetic efficiency is better under MoDS than under MiDS, in Arabidopsis YL [21,22], and plants that were described to be tolerant to SDS did not perform better under MiDS [18]. Exploring the molecular and physiological strategies that plants apply to cope with MiDS or MoDS is therefore essential to ensure our future agricultural productivity [17].

Photosynthetic ability is very essential, as it directly contributes to plant growth and productivity under DS [15,23]. Dealing with the negative effects of DS on growth and productivity will demand to evaluate the way it impacts photosynthesis and to understand plants′ responses properly, thus leaf photosynthesis analysis remains pivotal [4]. Additionally, it is well-defined that plants practice multiple stress situations that act together under natural field conditions and, in the case of DS, there is at least the need to consider the interaction with the light conditions [4,24]. 

Plants must retain a balance between the capture of light energy, its supply to the reaction centers, the production of NADPH and ATP and the utilization of these products for CO_2_ fixation and biosynthesis [25,26,27]. Plants, in response to DS, close their stomata to decrease water loss, which results in lower CO_2_ entry into the leaf and lower CO_2_ fixation, thus in lower need for NADPH and ATP [28,29]. In some cases, under MiDS and low light (LL) conditions, photosynthetic activity and, particularly, electron transport rate (ETR) and NADP^+^ reduction are preserved, but under high light (HL) conditions, an imbalance between light energy capture and photochemical energy use appears, leading to a decrease in ETR, which leads to a high level of energy dissipation as heat to prevent the formation of reactive oxygen species (ROS) [29,30,31]. Consequently, under DS the absorbed light energy exceeds what it can be used and, thus, it can damage the photosynthetic apparatus, with photosystem II (PSII) being particularly exposed to damage [13,32,33,34,35]. The light-harvesting, excitation transfer, charge separation and electron transfer in PSII are the essential reactions of photosynthesis and, consequently, principally regulate its total efficiency [36]. In general, overexcitation of PSII is prevented largely by dissipation of excess excitation energy as heat, a process that is called non-photochemical exciton quenching (NPQ), and is typically measured by chlorophyll *a* fluorescence quenching [37,38,39]. If this excess excitation energy is not quenched by NPQ, increased production of ROS occurs that can lead to oxidative stress and lipid peroxidation [37,40,41]. Lipid peroxidation can be assessed by malondialdehyde (MDA), that is widely recognized as a marker of oxidative stress [42,43,44].

Flavonoids are representative plant secondary products with at least 54 different molecules in the model plant *Arabidopsis thaliana* (35 flavonols, 11 anthocyanins and 8 proanthocyanidins) [45]. Flavonoids, as well as many other plant polyphenols, possess a chemical structure ideal for free radical scavenging [45,46]. Their antioxidant properties include reactivity to a variety of ROS [47,48,49,50,51,52], as well as metal chelating [53,54]. Drought stress enhances flavonoid accumulation [55,56], which, in turn, improves antioxidant capacity by reducing ROS and improving drought tolerance [57,58,59].

Responses to DS of *Arabidopsis thaliana* have commonly been studied in the Columbia accession, which is mainly used in plant research [17]. However, the reactions to DS are complicated and our understanding of the responses that contribute to sustaining plant growth and development during mild drought stress (MiDS), or moderate drought stress (MoDS), is incomplete [17,20].

Despite several investigates implicating flavonoids in plant drought responses, how the regulation of stress tolerance is related to leaf development stage is largely unknown. To address this issue, we investigated how flavonoid accumulation and lipid peroxidation are influenced in young and mature *Arabidopsis thaliana* leaves, under MiDS and MoDS, and how flavonoid accumulation modulates oxidative stress influencing PSII photochemistry in both leaf types in response to DS.

## 2. Results

### 2.1. Soil and Leaf Water Status under Drought Stress

Water deficit stress was induced gradually by withholding water [21] in a randomized block design with three different watering regimes, as follows: at the onset of drought stress (OnDS, watering stopped 24 h before sampling), soil water status, indicating the level of soil water stress, was 95–96% of soil capacity, at mild drought stress (MiDS, watering stopped six days before sampling), the soil water status was 66–68% of soil capacity and at moderate drought stress (MoDS, watering stopped ten days before sampling) the soil water status was 50–52% of soil capacity [22].

Leaf water status did not differ between mature leaves (ML) and young leaves (YL) at the OnDS (Figure 1a), but at MiDS and MoDS, YL retained significantly higher leaf water content than ML (Figure 1a).

### 2.2. Young Leaves Accumulated Greater Amounts of Flavonoids and Less MDA under Drought Stress

At the OnDS, there was no difference between ML and YL in flavonoid content, but, at MiDS and MoDS, YL accumulated a significantly greater amount of flavonoids than ML (Figure 1b). Young leaves at MiDS accumulated the same amount of flavonoids as ML at MoDS (Figure 1b).

The level of lipid peroxidation in ML and YL during DS treatments was assessed by malondialdehyde (MDA) content that was determined by the reaction with 2-thiobarbituric acid (TBA). Even during OnDS, YL possessed lower level of oxidative stress compared to ML, as evident by the lower level of lipid peroxidation (Figure 2a). In addition, during MiDS and MoDS the level of lipid peroxidation was lower in YL compared to ML (Figure 2a). 

### 2.3. Excess Excitation Energy in Young and Mature Leaves under Drought Stress

The level of excess excitation energy (EXC), estimated as *F_v_*′/*F_m_*′ × (1 − q*_p_*) [60], was lower at the OnDS in YL, compared to ML, and remained lower in also YL, during MiDS and MoDS (Figure 2b). Young leaves at MoDS managed to possess an even lower level of EXC than YL at MiDS (Figure 2b). The low level of EXC of YL at MoDS was at the same level as the EXC of YL at the OnDS (Figure 2b).

### 2.4. Light Energy Utilization in Photosystem II and Excess Excitation Pressure of Young and Mature Leaves 

The effective quantum yield of photochemistry (Φ*_PSII_*) in mature leaves (ML) decreased more at mild drought stress (MiDS) than at moderate drought stress (MoDS) (Figure 3a). These decreases were not compensated by the regulated energy dissipated in PSII (Φ*_NPQ_*) (Figure 3b), thus MiDS-ed ML showed the highest values of non-regulated energy dissipated in PSII (Φ*_NO_*), followed by MoDS-ed ML (Figure 4a). MoDS-ed YL showed higher Φ*_PSII_* values even from the onset of drought stress (OnDS) ML and from the MiDS-ed YL (Figure 3a). The highest Φ*_PSII_* values were recorded at all light intensities at the OnDS-ed YL (Figure 3a), accompanied by the highest Φ*_NPQ_* values at all light intensities (Figure 3b), thus presenting the lowest level of non-regulated energy dissipated in PSII (Φ*_NO_*) (Figure 4a). Φ*_NO_* values of MoDS-ed YL were lower from those of MiDS-ed YL that have the same Φ*_NO_* as the OnDS ML (Figure 4a). The highest excitation pressure (1 − q*_p_*) was recorded at MiDS-ed ML, followed by MoDS-ed ML and MiDS-ed YL (Figure 4b). The lowest 1 − q_*p*_ values were recorded at the OnDS-ed YL, followed by MoDS-ed YL (Figure 4b).

Mature leaves (ML) at the onset of DS and at MoDS, at light intensities up to 300 µmol photons m^−2^ s^−1^, showed the same quantum yield of photochemistry (Φ*_PSII_*) (Figure 3a), but those at the OnDS were more capable of dissipating the absorbed light energy that was not used for photochemistry as heat (Φ*_NPQ_*), than that of MoDS ML (Figure 3b), resulting in lower Φ*_NO_* values of the OnDS-ed ML (Figure 4a). Light intensities above 300 µmol photons m^−2^ s^−1^, that acted synergistically with the increased severity of DS, decreased the fraction of light energy that was used for photochemistry of MoDS-ed ML, compared to the OnDS-ed ML (Figure 3a), but stimulated their capability to dissipate the excess light (Figure 2b) as heat (Φ*_NPQ_*, Figure 3b). However, it seems that the increased Φ*_NPQ_* was not sufficient to keep the fraction of non-regulated energy dissipated in PSII (Φ*_NO_*) of MoDS-ed ML at a lower level than that of the OnDS-ed ML (Figure 4a), thus resulting in excess excitation pressure (1 − q*_p_*) of MoDS-ed ML (Figure 4b).

### 2.5. Correlation Analysis in Light Energy Utilization in Photosystem II and Lipid Peroxidation of Young and Mature Leaves under Drought Stress

The maximum efficiency of PSII photochemistry (*F_v_*/*F_m_*), at the three DS treatments, in both YL and ML, was significantly negatively correlated to the level of lipid peroxidation, measured as malondialdehyde (MDA) (Figure 5a). The decline in *F_v_*/*F_m_* is also showing the degree of PSII photoinhibition [43]. The level of excitation pressure (1 − q*_p_*), estimated at the growth light intensity of 136 µmol photons m^−2^ s^−1^ was strongly correlated at the three DS treatments in both YL and ML, to the level of MDA (Figure 5b). In addition, the maximum efficiency of PSII photochemistry (*F_v_*/*F_m_*) was significantly negatively correlated, at the three DS treatments in both YL and ML, to the level of excitation pressure (1 − q*_p_*) at the growth light intensity of 136 µmol photons m^−2^ s^−1^ (Figure 5c). The effective quantum yield of PSII photochemistry (Φ*_PSII_*), at the growth light intensity of 136 µmol photons m^−2^ s^−1^, was also significantly negatively correlated, at the three DS treatments in both YL and ML, to the level of 1 − q*_p_* (Figure 5d).

The quantum yield of non-regulated energy dissipated in PSII (Φ*_NO_*) was significantly positively correlated, at the three DS treatments, in both YL and ML, to both the level of MDA (Figure 6a) and the level of excitation pressure (1 − q*_p_*) (Figure 6b).

### 2.6. Correlation Analysis between Leaf Water Content and Flavonoid Accumulation of Young and Mature Leaves under Drought Stress 

The leaf water content (LWC) expressed as percentage of control (well-watered plants), at the three DS treatments, in both YL and ML, was significantly negatively correlated to flavonoid accumulation, also expressed as percentage of control (Appendix A).

## 3. Discussion

Under most environmental stresses, the absorbed light energy exceeds what it can be used, resulting in ROS generation, such as superoxide anion radical (O_2_**^•^****^−^**), hydrogen peroxide (H_2_O_2_) and ^1^O_2_ [38,41,61,62,63,64,65,66]. When ROS production is not counterbalanced by the antioxidant defense network, photo-oxidative stress occurs [24,41,61,62]. ROS–antioxidant interaction provides important knowledge for the redox state that impacts gene expression associated with plant stress responses modulating the initiation of photosynthetic acclimation or cell death [66,67,68,69,70]. 

The higher flavonoid accumulation of YL at MiDS than ML at the OnDS (Figure 1b) resulted in their ability, at PAR higher than 300 μmol m^−2^ s^−1^, to maintain lower ^1^O_2_ generation, as observed by the lower Φ*_NO_* values (Figure 4a). Φ*_NO_* consists of chlorophyll fluorescence interior conversions and intersystem crossing, which leads to the generation of ^1^O_2_ via the triplet state of chlorophyll (^3^chl^*^) that reacts with oxygen (O_2_) [71,72,73,74,75,76,77]. Thus, the lower Φ*_NO_* values suggest a lower level of ^1^O_2_ formation in YL, compared to ML. Singlet oxygen is a highly damaging ROS created in PSII [71,75,76,77] and high concentrations of ^1^O_2_ activate programmed cell death [69,70]. Accumulation of excess excitation energy (EXC) may lead to the proportional increase in the production of ^1^O_2_, that can cause specific damage [78]. In accordance to this, the light response curves of excitation pressure (1 − q*_p_*) and of the non-regulated energy dissipated in PSII (Φ*_NO_*) show a similar trend (Figure 4), with a significant positive correlation to be observed (Figure 6b). Excitation pressure (1 − q*_p_*) was also strongly correlated with the level of lipid peroxidation (Figure 5b), that was also strongly correlated with Φ*_NO_* (Figure 6a). 

While scavenging of O_2_**^•−^** is mainly achieved by the antioxidant enzyme, superoxide dismutase [41,79], ^1^O_2_ can only be controlled by non-enzymatic antioxidants [51]. Singlet oxygen is quenched by the plant antioxidants β-carotene, α-tocopherol, plastoquinones, zeaxanthin and flavonoids [51,71,72,77]. However, ROS produced in chloroplasts are not only creating oxidative stress but also confer significant biological functions, such as redox signaling in the regulation of leaf development and translating information from the environment [69,70,74,80,81,82]. NPQ has also been suggested to be involved in the mechanism of plant acclimation to biotic or abiotic stress and to be a major component of the systemic acquired resistance [66,70,83,84,85,86].

Mature leaves maintained higher PSII photochemistry (Φ*_PSII_*) at MoDS than YL at MiDS at low PAR (<300 μmol m^−2^ s^−1^) (Figure 3a), but YL, by dissipating slightly more absorbed light energy as heat (Φ*_NPQ_*) (Figure 3b), retained lower singlet oxygen (^1^O_2_) generation, as observed by lower Φ*_NO_* (Figure 4a). An increase in the values of Φ*_PSII_* and a decrease in the values of Φ*_NPQ_* is observed when the leaf gets older [87]. In young leaves, only a fraction of absorbed light energy is utilized in photochemistry via CO_2_ assimilation, because carbon assimilation capacity is developed later than light capture ability [88,89,90,91], thus they have to activate NPQ to dissipate excessive excited energy as heat and regulate photosynthetic electron flow during photosynthetic induction [92,93] in order to avoid the harmful generation of ^1^O_2_ that can damage the photosynthetic apparatus [37,91,94]. Thus, YL, by regulating NPQ, can maintain a balanced ROS level that allows growth [69,70,85,86,95] and prevents their oxidative damage [22]. Dissipation of excess excitation energy in YL plays an important role in order to avoid possible photodamage to PSII under DS conditions [91]. PSII photodamage is caused by ROS produced by excess excitation or/and by other photosensitizers, primarily the Mn cluster [31].

Drought stress leads to decreases in the fraction of open reaction centers of PSII (q*_p_*), reductions in the effective quantum yield of photochemistry (Φ*_PSII_*) and increases in the regulated energy dissipated in PSII (Φ*_NPQ_*) and these changes become bigger with stress duration [15,21,22,96,97]. Photochemical quenching of chlorophyll *a* fluorescence (q*_p_*) estimates changes in the redox state of Q_A_ and, thus, the reduction level of PSII reaction centers, while 1 − q*_p_*, the fraction of reduced quinone Q_A_, illustrates the degree of excitation pressure on PSII [34,98]. The higher 1 − q*_p_*, the higher the excitation pressure [44,99,100]. Mature leaves at MoDS and YL at MiDS accumulated the same level of flavonoids (Figure 1b) and experienced the same level of lipid peroxidation (Figure 2a), exhibiting the same level of excitation pressure (1 − q*_p_*) (Figure 4b).

MoDS *A. thaliana* leaves had a significantly higher flavonoid accumulation that serves as a sufficient antioxidant mechanism destroying ROS [51,58]. As LWC decreased, flavonoid accumulation increased and a negative significant correlation between LWC and flavonoids was observed (Appendix A). Thus, flavonoid accumulation may have an important role in the acclimation process to DS. Flavonoid accumulation, in YL under MiDS and in ML under MoDS (Figure 1b), was not enough to prevent ^1^O_2_ generation, as observed by Φ*_NO_* values, but it appears that it was sufficient in YL under MoDS (Figure 1b), that resulted in reduced excess light availability, similar to the level of the OnDS-ed YL (Figure 2b), with only slightly higher excitation pressure from them (Figure 4b).

Chlorophyll fluorescence analysis has been widely used to acquire knowledge about the function of the photosynthetic machinery and for the assessment of photosynthetic tolerance mechanisms to biotic [66,86,101,102] and abiotic stresses [103,104,105,106,107,108], including drought stress [21,22,96,108,109,110]. However, DS may not affect a plant leaf uniformly [97], thus photosynthetic performance may be extremely heterogeneous at the leaf surface, denoting conventional chlorophyll fluorescence measurements non-characteristic of the physiological status of the entire leaf [111,112,113]. This disadvantage overcomes chlorophyll fluorescence imaging analysis, which permits the detection of spatiotemporal heterogeneity at the total leaf surface [114].

Among the chlorophyll fluorescence parameters used for DS monitoring, evaluation and selection of drought-tolerant species, decreases in the maximum efficiency of PSII photochemistry (*F_v_*/*F_m_*) were mostly used [109]. However, there are research results that question the efficacy of *F_v_*/*F_m_* as a good indicator of DS [109,115,116]. Recently, the reduction status of the plastoquinone pool, or, in other words, PSII excitation pressure, was found to be the most sensitive and appropriate indicator to probe photosynthetic function and determine the impact of biotic and abiotic stresses on leaf photosynthesis [66,117]. Our results collaborate this opinion, suggesting the use of PSII excitation pressure (1 − q*_p_*) as a good indicator to reveal short- or long-term stress impact on the mechanisms of PSII functionality.

Drought stress enhances flavonoid accumulation [55,56], which, in turn, can reduce ROS accumulation, improving the antioxidant capacity and, consequently, resulting in drought acclimation [57,58,59]. Flavonoid production is one of the strategies used by native species living in extreme environments to avoid the oxidative damage caused by drought [118,119]. Our study indicates that higher concentrations of flavonoids in YL, compared to ML (Figure 1b), promote acclimation to MoDS of YL by helping to prevent ^1^O_2_ generation, as observed by the lower Φ*_NO_* values in YL, compared to ML (Figure 4a). Differences in drought tolerance between YL and ML of *Arabidopsis thaliana* reflect their ability to respond to oxidative stress by increasing flavonoid accumulation. 

Overall, acclimation of YL to MoDS (and not to MiDS) was correlated with higher flavonoid accumulation, decreased lipid peroxidation, higher PSII photochemistry (Φ*_PSII_*) than ML leaves, lower excitation pressure (1 − q*_p_*), lower excess excitation energy (EXC) and also a higher capacity to maintain low Φ*_NO_*, which can be effectively used for selecting drought tolerant plants. Breeding of plants with high flavonoid content that confers drought resistance and resilience could help crop production under future climate change [78].

## 4. Materials and Methods

### 4.1. Plant Material, Growth Conditions and Drought Stress Treatment 

*Arabidopsis thaliana* ecotype Columbia (Col-0) seedlings were grown in a growth chamber with controlled environmental conditions, under a long day photoperiod of 14 h/10 h, with 40 ± 5/55 ± 5 % day/night humidity, temperature of 22 ± 1/19 ± 1 °C day/night and light intensity of 130 ± 10 μmol photons m^−2^ s^−1^. Drought stress was imposed by withholding water for a period up to 10 days on 4-week-old Arabidopsis plants [14]. The two developmental leaf stages that were studied were fully developed mature leaves (ML) and developing young leaves (YL) from plants whose watering stopped twenty-four hours before sampling and characterized as onset of drought stress (OnDS), six days before sampling, characterized as mild drought stress (MiDS), and ten days before sampling, characterized as moderate drought stress (MoDS), representing three categories of drought stressed (DS) plants. As young leaves were considered those in the middle of the leaf rosette with 1.5–2 cm length, while the typical length of mature leaves in the rosette was 4.1 ± 0.5 cm [21,22].

### 4.2. Soil and Leaf Water Status

Soil volumetric water content (SWC) in m^3^ m^−3^ was measured as described previously [91] with a 5TE (Decagon Devices, Pullman, WA, USA) soil moisture sensor that uses a two-sensor array to measure electrical conductivity, coupled to the read-out device ProCheck (Decagon Devices, Pullman, WA, USA). 

Plant leaf water status was determined by measuring the leaf water content (LWC) by the electronic moisture balance (MOC- 120H, Shimadzu, Tokyo, Japan) using the formula: (FW − DW)/DW × 100%, where FW is the fresh weight and DW refers to dry weight [21].

### 4.3. Lipid Peroxidation Measurements

Lipid peroxidation was measured as malondialdehyde (MDA) content determined by reaction with 2-thiobarbituric acid (TBA) [120]. The concentration of MDA was calculated from absorbance read at 440 nm, 532 nm and 600 nm spectrophotometrically (PharmaSpec UV-1700; Shimadzu, Tokyo, Japan) as follows: [(Abs 532_+TBA_) − (Abs 600_+TBA_) − (Abs 532_−TBA_−Abs 600_−TBA_)] = A
[(Abs 440_+TBA_ − Abs 600_+ TBA_) 0.0571] = B
where 532 nm is the maximum absorbance of the TBA-MDA complexes, 600 nm is the correction factor for nonspecific turbidity and 440 nm is the correction factor for sucrose interference. 

Finally, MDA equivalents were calculated as:MDA equivalents (nmol mL^−1^) = (A − B)/157,000 × 10^6^
where 157,000 is the molar extinction coefficient for MDA.

### 4.4. Determination of Flavonoids

Leaf discs (1cm in diameter), from each treatment of *A. thaliana* mature leaves (ML) and young leaves (YL), from control plants and DS plants, were ground into fine powder in liquid nitrogen. For flavonoid determination, the frozen powder was extracted in 10 cm^3^ of acidified methanol (HCl:methanol, 1:99, *v/v*), as described by Havaux and Kloppstech [121]. Absorption spectra of the extracts were determined after centrifugation at 5000× *g* for 10 min, using a PharmaSpec UV-1700 spectrophotometer (Shimadzu, Tokyo, Japan). Absorbance was read at 350 nm and flavonoid content was expressed as absorbance cm^−2^ [121].

### 4.5. Chlorophyll Fluorescence Analysis

Chlorophyll fluorescence was measured in dark-adapted *A. thaliana* young and mature leaves using an imaging-PAM fluorometer (Walz, Effeltrich, Germany), as described previously [91]. Light curves were used for the calculation of various fluorescence parameters at photosynthetic photon flux density (PPFD) of 0, 6, 46, 136, 226, 336, 436, 636, 736, 894, 1011, 1211 and 1386 μmol photons m^−2^ s^−1^. The chlorophyll fluorescence parameters measured were the minimum chlorophyll *a* fluorescence in the dark (*Fo*), the maximum chlorophyll *a* fluorescence in the dark (*F_m_*), the maximum chlorophyll *a* fluorescence in the light (*F_m_*′) and the steady-state photosynthesis in the light (*F_s_*). The minimum chlorophyll *a* fluorescence in the light was calculated by the Imaging Win V2.41a software (Heinz Walz GmbH, Effeltrich, Germany) as *Fo*′ = *Fo*/(*F_v_*/*F_m_* + *Fo*/*F_m_*′) [122]. The maximum efficiency of PSII photochemistry (*F_v_*/*F_m_*, where *F_v_* = *F_m_* − *Fo*) [123], the effective quantum yield of PSII photochemistry (Φ*_PSII_* = [*F_m_*′ − *F_s_*]/*F_m_*′) [124,125], the quantum yield of regulated non-photochemical energy loss in PSII (Φ*_NPQ_* = *F_s_*/*F_m_*′ − *F_s_*/*F_m_*) [126] and the quantum yield of non-regulated energy dissipated in PSII (Φ*_NO_* = *F_s_*/*F_m_*) [126] were calculated. We also measured the relative PSII electron transport rate (ETR = Φ*_PSII_* × PAR × c × abs, where PAR is the photosynthetically active radiation, c is 0.5 and abs is the total light absorption of the leaf taken as 0.84), the proportion of closed PSII reaction centers, referred to as excitation pressure and calculated as 1 − q*_p_* [98], where q*_p_* = [*F_m_*′ − *F_s_*]/[*F_m_*′ − *Fo*′], the non-photochemical quenching, that reflects heat dissipation of excitation energy (NPQ = [*F_m_* − *F_m_*′]/*F_m_*′ [127], and the excess excitation energy (EXC = *F_v_*′/*F_m_*′ × (1 − q*_p_*) [60].

A linear regression analysis was also performed [14].

### 4.6. Statistical Analysis

Each treatment was analyzed with five–six replicates from five–six different leaves from different plants. A standard error (SE) was calculated and data were expressed as mean ± SE (n = 5–6). Statistically significant differences between the treatments were analyzed by analysis of variance (ANOVA) using the software StatView (SAS Institute, Cary, NC, USA) [22].

## Figures and Tables

**Figure 1 molecules-26-04157-f001:**
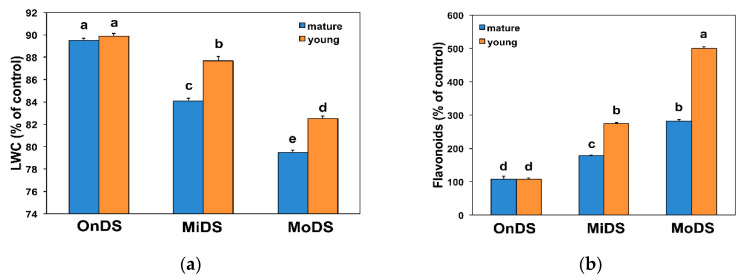
The relative leaf water content (**a**) and the flavonoid content (**b**) expressed as percentage of control (well-watered plants), in *A. thaliana* young and mature leaves at the onset of drought stress (OnDS), at mild drought stress (MiDS) and at moderate drought stress (MoDS). Error bars represent ± standard error of the mean (n = 5–6). Bars with different lowercase letters are significantly different at *p* < 0.05.

**Figure 2 molecules-26-04157-f002:**
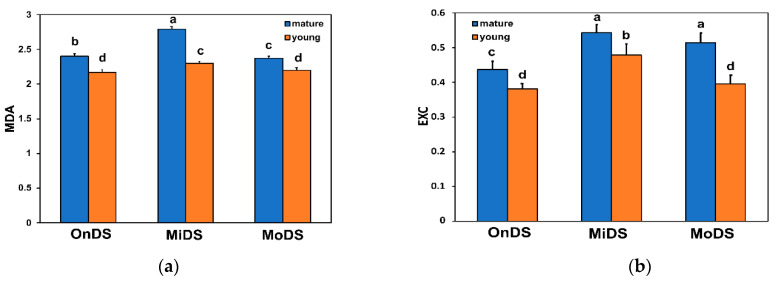
Changes in the level of lipid peroxidation, measured as nmol MDA g^−1^ fresh weight (**a**) and in the excess excitation energy (EXC), estimated according to Demmig-Adams et al. [60] as: *F_v_*′/*F_m_*′ × (1 − q*_p_*) (**b**), in *A. thaliana* young and mature leaves at the onset of drought stress (OnDS), at mild drought stress (MiDS) and at moderate drought stress (MoDS). Error bars represent ± standard error of the mean (n = 5–6). Bars with different lowercase letters are significantly different at *p* < 0.05.

**Figure 3 molecules-26-04157-f003:**
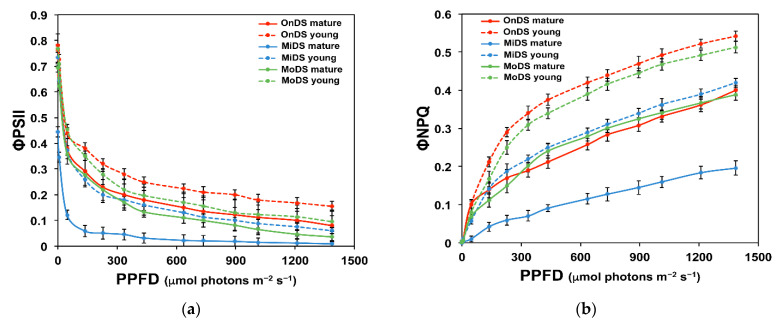
Light response curves of the effective quantum yield of photochemistry (Φ*_PSII_*) (**a**) and of the regulated energy dissipated in PSII (Φ*_NPQ_*) (**b**), in *A. thaliana* young and mature leaves, at the onset of drought stress (OnDS), at mild drought stress (MiDS) and at moderate drought stress (MoDS). Error bars represent ± standard error of the mean (n = 5–6).

**Figure 4 molecules-26-04157-f004:**
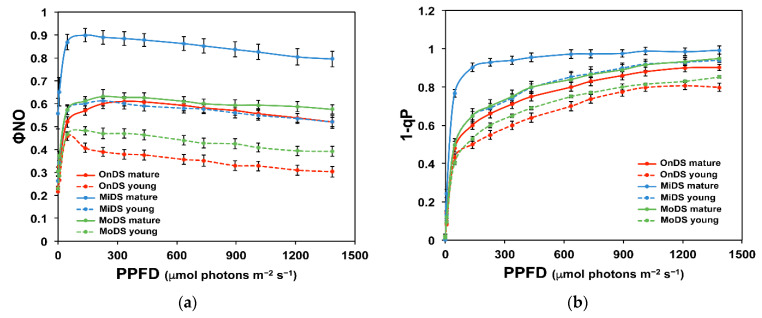
Light response curves of non-regulated energy dissipated in PSII (Φ*_NO_*) (**a**) and of excitation pressure (1 − q*_p_*) (**b**), in *A. thaliana* young and mature leaves at the onset of drought stress (OnDS), at mild drought stress (MiDS) and at moderate drought stress (MoDS). Error bars represent ± standard error of the mean (n = 5–6).

**Figure 5 molecules-26-04157-f005:**
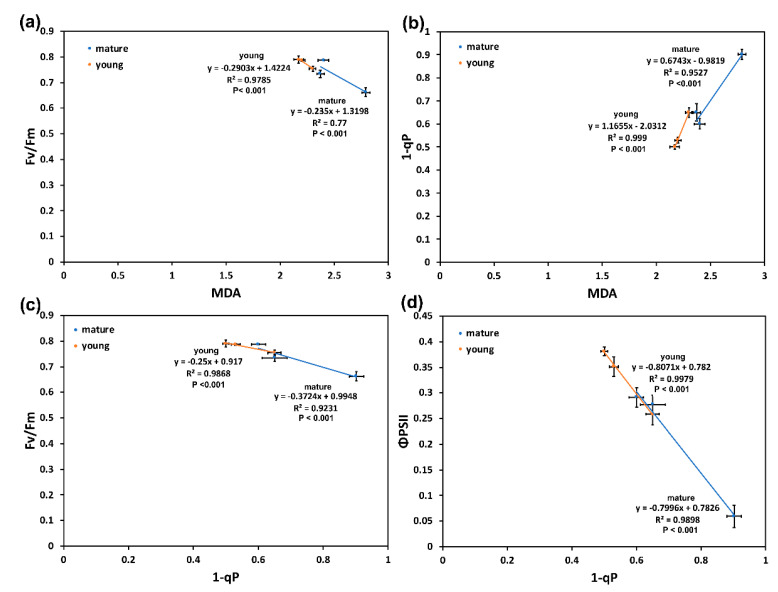
The relationship between the maximum efficiency of PSII photochemistry (*F_v_*/*F_m_*) and the level of lipid peroxidation, measured as malondialdehyde (MDA) (**a**), the level of MDA with the level of excitation pressure (1 − q*_p_*) (**b**), the level of excitation pressure (1 − q*_p_*) with *F_v_*/*F_m_* (**c**) and the effective quantum yield of PSII photochemistry (Φ*_PSII_*) with 1 − q*_p_* (**d**), in both *A. thaliana* young leaves (YL) and mature leaves (ML) at the three DS treatments. Error bars represent ± standard error of the mean (n = 5–6).

**Figure 6 molecules-26-04157-f006:**
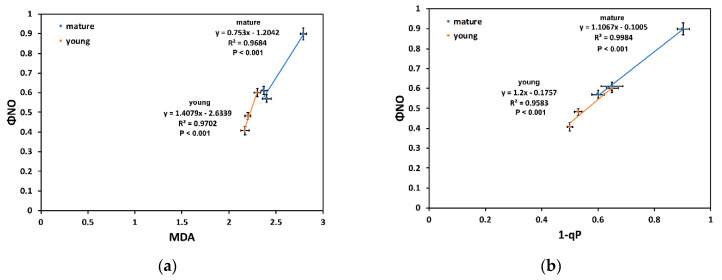
The relationship between the quantum yield of non-regulated energy dissipated in PSII (Φ*_NO_*) and the level of MDA) (**a**) and the level of excitation pressure (1 − q*_p_*) with Φ*_NO_* (**b**), in both YL and ML, at the three DS treatments. Error bars represent ± standard error of the mean (n = 5–6).

## Data Availability

The data presented in this study are available in this article and in Appendix A.

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
