# Peer review of "Leaf Age-Dependent Photosystem II Photochemistry and Oxidative Stress Responses to Drought Stress in Arabidopsis thaliana Are Modulated by Flavonoid Accumulation"

_molecules, 2021, doi:10.3390/molecules26144157_

Round 1
Reviewer 1 Report
The article under review aims to establish a correlation between the drought stress and flavonoid accumulation in Arabidopsis thaliana leaves having different developmental stages. The topic is interesting as it aims contributing to unveil the in planta function of flavonoids and offer a better understanding of plant adaptation against drought stress. The abstract is well written and concise, the introduction provides an updated background of the topic. Material and methods are clearly and reproducibly described, the authors use standard research methods for the field.
Results are clear and well illustrated. Conclusions are supported by experimental evidence and highlight the importance of the study.
Author Response
The article under review aims to establish a correlation between the drought stress and flavonoid accumulation in Arabidopsis thaliana leaves having different developmental stages. The topic is interesting as it aims contributing to unveil the in planta function of flavonoids and offer a better understanding of plant adaptation against drought stress. The abstract is well written and concise, the introduction provides an updated background of the topic. Material and methods are clearly and reproducibly described, the authors use standard research methods for the field.
Results are clear and well illustrated. Conclusions are supported by experimental evidence and highlight the importance of the study.
Thank you for your positive comments.
Reviewer 2 Report
The paper “"Leaf Age-Dependent Photosystem II Photochemistry and Oxi-dative Stress Responses to Drought Stress in Arabidopsis thaliana are correlated to Flavonoid Accumulation " proposed for publication in " Molecules" aims to investigate the antioxidant and protective effects of Flavonoid against drought in young and mature leaves from Arabidopsis thaliana. Such effects of flavonoids are already assessed, and the novelty is given by the comparison of data from leaves at two growing stages. The text is well written. However, the procedures and data treatment need to be better clarified. In fact, while the title promises a correlation study of age-dependent photosystem II photochemistry and oxi-2 dative stress responses to flavonoid accumulation, a direct correlation between these parameters is not presented. Therefore, a revision is required to improve the text, as specified below
Major remarks:
- The title promises a correlation study of age-dependent photosystem II photochemistry and oxidative stress responses vs flavonoid accumulation. However, the actual correlation studies are between different parameters of photosystem II photochemistry and functionality, or between each of these parameters and MDA, but not vs flavonoids. Is it kindly possible to provide also this topical correlation?
- Different assays have been performed on the leaves, in order to assess a relationship between drought, oxidative effects (lipid peroxidation), PSII functionality parameters, and flavonoids. A breakdown presentation should be given to make immediate and clear to the reader how measurements on leaves and collection of samples for biochemistry were performed. In other words, if samples were taken on the same leaf used to measure fluorescence, and if the samples indicated as 5-6 for each data mean were taken from the same of from different leaves.
- If possible, correlations should be given by using couples of single measurement data taken from the same leaf for each kind of treatment, instead of means.
Minor remark:
Point 4.4 – Given the detailed description of MDA measurement procedure, please describe in more detail also the determination of flavonoids.
In bar chart Figures (1,2) statistics significance should be indicated with asterisks and horizontal brackets above placed significantly different data, instead of less common letters.
Author Response
The paper “"Leaf Age-Dependent Photosystem II Photochemistry and Oxidative Stress Responses to Drought Stress in Arabidopsis thaliana are correlated to Flavonoid Accumulation " proposed for publication in " Molecules" aims to investigate the antioxidant and protective effects of Flavonoid against drought in young and mature leaves from Arabidopsis thaliana. Such effects of flavonoids are already assessed, and the novelty is given by the comparison of data from leaves at two growing stages. The text is well written. However, the procedures and data treatment need to be better clarified. In fact, while the title promises a correlation study of age-dependent photosystem II photochemistry and oxidative stress responses to flavonoid accumulation, a direct correlation between these parameters is not presented. Therefore, a revision is required to improve the text, as specified below
Thank you for your comments that helped us to improve our manuscript.
Major remarks:
- The title promises a correlation study of age-dependent photosystem II photochemistry and oxidative stress responses vs flavonoid accumulation. However, the actual correlation studies are between different parameters of photosystem II photochemistry and functionality, or between each of these parameters and MDA, but not vs flavonoids. Is it kindly possible to provide also this topical correlation?
You are right, the title was promising a correlation study that was not provided, since flavonoid accumulation was in a dose dependent response to drought but lipid peroxidation and PSII photochemistry in Arabidopsis thaliana did not show a dose dependent response (Abstract L. 30). Thus, we changed the title accordingly to “Leaf Age-Dependent Photosystem II Photochemistry and Oxidative Stress Responses to Drought Stress in Arabidopsis thaliana are Modulated by Flavonoid Accumulation”. Thank you for pointing this. We added also (lines 201-205) correlation analysis of flavonoid accumulation with leaf water content that both show a dose dependent response and were strongly negative correlated (supplemental Figure 1) and we discuss it in lines 267-271.
- Different assays have been performed on the leaves, in order to assess a relationship between drought, oxidative effects (lipid peroxidation), PSII functionality parameters, and flavonoids. A breakdown presentation should be given to make immediate and clear to the reader how measurements on leaves and collection of samples for biochemistry were performed. In other words, if samples were taken on the same leaf used to measure fluorescence, and if the samples indicated as 5-6 for each data mean were taken from the same of from different leaves.
More details on how measurements on leaves and collection of samples were done are presented now in the section Materials and Methods (lines 381-382). Samples were taken from 5-6 different leaves from different plants. Samples e.g., for lipid peroxidation measurements were not taken from the same leaf used to measure fluorescence, but from different leaves, because mechanical wounding during the process of measuring chlorophyll fluorescence might have increase MDA content.
- If possible, correlations should be given by using couples of single measurement data taken from the same leaf for each kind of treatment, instead of means.
We will try to apply your suggestion in our next work. In this work the couples of single measurements were not taken from the same leaf that used to measure fluorescence, e.g., leaf water content, flavonoid accumulation or lipid peroxidation were not from the same leaf that used to measure fluorescence. Only the different chlorophyll fluorescence parameters were from the same leaf for each kind of treatment.
Minor remark:
Point 4.4 – Given the detailed description of MDA measurement procedure, please describe in more detail also the determination of flavonoids.
Determination of flavonoids was given in more detail (lines 350-357).
In bar chart Figures (1,2) statistics significance should be indicated with asterisks and horizontal brackets above placed significantly different data, instead of less common letters.
The use of horizontal brackets with asterisks in some cases is explanatory but in other cases is confusing. By using different lowercase letters, we show all combination significant comparisons.